# Immature Sword Bean (*Canavalia gladiata*) Pod Alleviates Allergic Rhinitis (A Double-Blind Trial) Through PI3K/Akt/mTOR Signaling

**DOI:** 10.3390/nu17030468

**Published:** 2025-01-28

**Authors:** Hye-Jeong Hwang, Hyeock Yoon, Joo-Hyung Cho, Seong Lee, Kyung-A Hwang, Young Jun Kim

**Affiliations:** 1Department of Agrofood Resources, National Institute of Agricultural Sciences, Rural Development Administration, Wanju-gun 55365, Republic of Korea; hjh1027@korea.kr; 2Department of Food and Biotechnology, Korea University, Sejong-si 30019, Republic of Korea; saranghe11@korea.ac.kr; 3Myongji Bioefficacy Research Center, Myongji University, Yongin-si 17058, Republic of Korea; chovincent@mju.ac.kr; 4Biomedical Research Institute, Dankook University Hospital, Cheonan-si 31116, Republic of Korea; seonglee@empas.com

**Keywords:** immature sword bean pod, *Canavalia gladiata*, allergic rhinitis, eosinophil cationic protein

## Abstract

**Background:** Allergic rhinitis is an IgE-mediated condition of nasal congestion, runny nose, and sneezing which significantly impairs the quality of life. Current treatments, including antihistamines, often have long-term side effects, leading patients to seek safer alternatives. **Objectives:** Therefore, in this study, we aimed to evaluate the symptom relief efficacy of immature sword bean pod (SBP) extract, a natural material, in patients with allergic rhinitis, explore the mechanisms by which SBP regulates allergic immune responses, and evaluate its efficacy and safety as a functional ingredient in the management of allergic rhinitis. **Methods:** In a double-blind, placebo-controlled, randomized trial involving 64 participants with perennial allergic rhinitis, the subjects were assigned to receive either SBP or placebo orally for six weeks. **Results:** The SBP group exhibited significant improvements in nasal congestion (interaction *p* = 0.031), RQLQ (interaction *p* = 0.001), sleep (interaction *p* = 0.004), systemic reaction (interaction *p* = 0.002), daily life (interaction *p* = 0.047), and nasal symptoms (interaction *p* = 0.002). SBP treatment in EoL-1 and HMC-1 cells also led to a notable reduction in eosinophil cationic protein levels (*p* < 0.05), a key biomarker of allergic inflammation, by inhibiting PI3K/Akt/mTOR activation, resulting in decreased eosinophil activity. **Conclusions:** These findings suggest that the SBP extract is a promising natural treatment for allergic rhinitis, offering both efficacy and safety by improving key symptoms and reducing inflammatory responses.

## 1. Introduction

Allergic rhinitis (AR) is an inflammatory response of the nasal mucosa that occurs upon exposure to specific allergens, leading to symptoms such as sneezing, nasal congestion, rhinorrhea, and nasal pruritus [1]. AR is classified as seasonal allergic rhinitis (SAR), which is triggered by allergens such as pollen and fine dust during specific seasons, and perennial allergic rhinitis (PAR), which is caused by allergens such as house dust mites and pet dander [2]. In particular, fine dust, a seasonal allergen, contains very small particulate matter that penetrates deeply into the human respiratory system, exacerbating symptoms of allergic rhinitis and potentially leading to various respiratory diseases, such as asthma, chronic obstructive pulmonary disease (COPD), and pneumonia [3,4]. In South Korea, AR is a prevalent condition, affecting an estimated 20–30% of the population, with six particularly high rates among children and adolescents. According to the Health Insurance Review and Assessment Service (HIRA Research), the number of patients receiving medical care for AR during the spring months (March to May) will increase from approximately 2.75 million in 2022 to approximately 3.81 million in 2023, marking a 38% year-over-year increase. In the fall season (September to November), the number of allergic rhinitis patients increased by 16%, from 3.4 million in 2022 to 3.95 million in 2023, indicating that there are more allergic rhinitis patients in the fall than there are in the spring [5]. AR treatment aims to relieve symptoms and avoid triggering factors. Recent advancements in AR treatment have focused on biological agents that suppress specific immune responses (monoclonal antibody therapies), novel antihistamines that enhance the efficacy while reducing the side effects of traditional medications, and personalized treatment plans based on genetic analyses to identify individual allergic responses [1,6]. Notably, there is growing interest among patients with AR in personalized health supplements formulated with natural ingredients based on individual health conditions and genetic information, offering fewer side effects and lower long-term use burdens. Consequently, there is a need to develop personalized treatments that utilize natural components to meet the demands of patients with AR.

Unlike healthy individuals, patients with allergies experience abnormal immune responses. In these patients, the number of granular white blood cells, known as eosinophils, increases. These eosinophils escape from the bone marrow, circulate in the blood, and target specific tissues [7]. Upon reaching the target inflammatory tissue, eosinophils are activated and secrete cytokines, inflammatory mediators, and eosinophil cationic proteins (ECPs). This secretion leads to epithelial damage, smooth muscle contraction, and recruitment of inflammatory cells, resulting in symptoms such as itching and swelling [8]. Eosinophils and ECP, secreted exclusively by eosinophils, are used to evaluate allergic asthma [9]. In particular, activated ECP can exert various biological effects, primarily by degranulating mast cells and releasing allergens, such as histamine and leukotrienes from mast cell granules [10,11]. Identifying the cellular sources and molecular mechanisms involved in ECP production that play a critical role in the pathophysiology of allergic asthma is essential.

Eosinophils that are activated by inflammatory and allergic responses and are closely associated with the phosphoinositide 3-kinase (PI3K)/protein kinase B (Akt)/mammalian target of rapamycin (mTOR) pathway [12]. Activation of this pathway enhances eosinophil survival and activation, leading to the degranulation of cytotoxic proteins, including ECP [13]. By promoting eosinophil activity, this pathway increases the production of inflammatory cytokines, such as interleukin-4 (IL-4) and interleukin-5 (IL-5), in conditions such as asthma and chronic rhinosinusitis with nasal polyps (CRSwNP), contributing to chronic inflammation [14]. In addition, the PI3K/Akt/mTOR pathway plays a key role in regulating the Th1/Th2 cell balance, which is critical in allergic responses. Previous studies have shown that PI3K inhibitors suppress pathological features associated with asthma, including chemokine secretion by activated eosinophils, increased levels of IL-5 and IL-13 in the bronchi, and eosinophil infiltration in lung tissue [15]. This evidence highlights the significant role of PI3K/Akt signaling in the pathogenesis of asthma. mTOR, a serine/threonine-specific protein kinase within the PI3K family, is central to regulating cell growth and metabolism [16]. In asthma, mTOR becomes activated through phosphorylation, disrupting the balance of Th17/Treg and Th1/Th2 cells [17]. This imbalance contributes to mast cell degranulation and promotes immune dysregulation and airway smooth muscle proliferation via the PI3K/mTOR pathway [18,19]. Therefore, the PI3K/Akt/mTOR pathway serves as a crucial regulator of eosinophil survival, activation, and degranulation, making it a potential therapeutic target for controlling eosinophil function and ECP release to thereby reduce tissue damage and alleviate the symptoms of inflammatory and allergic diseases.

The sword bean (*Canavalia gladiate*, SB) is a climbing annual plant belonging to the legume family and is primarily distributed in Southeast Asia, Africa, and the tropical regions of India [20]. Traditionally, sword beans have been used to treat inflammations such as sinusitis, rhinitis, gastritis, and colitis. It has also been used to promote blood circulation by resolving blood stasis and treating bronchial conditions such as throat paralysis and throat sores [21]. All parts of the SB are edible, including beans, pods, stems, and roots. However, non-bean parts are underutilized, and research investigating their functional properties is limited [22]. SB contains various bioactive compounds such as saponins, tannins, terpenoids, flavonoids, Canavalia gibberellins I and II, and steroids [23], with reported side effects that include anti-obesity [24], anti-inflammatory [25], antioxidant [26], hematopoietic improvement [27], and liver protection properties [23]. However, research examining these pods is limited. The pods of SB (SBP) can be consumed while still immature before the beans fully develop, and they have been studied for their anti-obesity [24], anti-inflammatory [28], and anti-allergic properties [21]. In our previous study on anti-allergic effects, SBP was demonstrated to suppress inflammatory cytokines and allergic mediators by regulating the PI3K/Akt/mTOR-signaling pathway in both cell and animal models [21].

Therefore, this study investigated the efficacy and mechanism of SBP in supplementation on hypersensitive immune allergies through clinical and in vitro experiments. Using this approach, our study aimed to establish the therapeutic potential and safety of SBP extract as a functional ingredient for alleviating allergic rhinitis symptoms.

## 2. Materials and Methods

### 2.1. Study Population

This was an observational study focused on the improvement of allergic rhinitis symptoms using SBP. This study was approved by the Institutional Review Board (IRB) of Dankook University Hospital (IRB NO. DKUH 2021-12-014, Date. 5 April 2022) and registered on Clinical Research Information Service (KCT0010091). The sample size was calculated using G-power 3.1.9.7 [29]. The calculated effective sample size was greater than 44 individuals, and considering a dropout rate of 15% and a compliance rate of 15%, the number of subjects registered in each group was greater than 28. Among the selected sample sizes, the effective sample size was 48; however, 64 individuals were selected to ensure a power of greater than 83%. Subjects with allergic rhinitis symptoms were recruited, and a double-blind, randomized, placebo-controlled clinical trial was conducted. The participants were divided into two groups, placebo and SBP, at a 1:1 ratio.

The inclusion criteria were as follows [30,31,32,33,34,35]: (1) aged 19 to 65 years; (2) WBC within the normal range (4–10 × 10^3^/μL) at the screening visit; (3) allergic nasal symptoms to allergens throughout the year within the past 12 months; (4) total nasal symptom score ≥ 4; (5) not receiving treatment for allergic rhinitis (drug therapy, non-drug therapy, or other treatments) during the trial period; and (6) agreed in writing to voluntarily participate in this examination and to comply with the requirements.

The exclusion criteria were as follows: (1) Subjects who used immunosuppressants, antibiotics, NSAIDs, steroids, or antihistamines within the last two weeks. (2) Subjects with autoimmune diseases, systemic inflammatory diseases, metabolic syndrome, infectious diseases, or severe cardiovascular disease under treatment. (3) Subjects with uncontrolled hypertension (≥160/100 mmHg) or who started new antihypertensive medication within the last three months. (4) Subjects with uncontrolled diabetes (fasting glucose ≥180 mg/dL) or who started new antidiabetic medication within the last three months. (5) Subjects with hepatic or renal dysfunction (AST, ALT, or γ-GTP ≥ 100 IU/L; creatinine ≥ 1.4 mg/dL). (6) Subjects with gastrointestinal conditions affecting absorption (e.g., Crohn’s disease, ulcers) or prior surgery, excluding simple appendectomy or hernia repair. (7) Subjects with a history of hypersensitivity to SBP or related drugs (e.g., aspirin, antibiotics). (8) Subjects receiving thyroid disease treatment. (9) Subjects diagnosed with or treated for cancer within the last year. (10) Subjects who had surgery within the last six months. (11) Subjects who took investigational products or SBP within the last three months. (12) Subjects who donated whole blood within 60 days, component blood within 30 days, or received a transfusion within 30 days of study initiation. (13) Subjects who used enzyme-inducing or inhibiting drugs (e.g., barbiturates) within 30 days of study initiation. (14) Subjects with excessive alcohol intake (men: ≥30 g/day; women: ≥20 g/day). (15) Subjects with a history of drug abuse or neuropsychiatric disorders (e.g., depression, schizophrenia). (16) Pregnant, breastfeeding, or postpartum women within six months. (17) Any other condition deemed unsuitable by the investigator.

### 2.2. Preparation of the SBP

SBP was sourced from Hwasun, Jeollanam-do, South Korea. To prepare the raw material, SBP was first washed with water to remove impurities, minced, and dried at 50 °C for 8 h (Cheil Machinery, Icheon, Republic of Korea). The dried material was then ground using an IKA M20 grinder (Staufen, Germany). For extraction, the SBP powder was mixed with 30% ethanol and stirred at 80 °C for 8 h. A secondary extraction was conducted under the same conditions (80 °C with 30% ethanol) for an additional 4 h. The combined extracts were filtered, and the filtrate was concentrated with a vacuum rotary evaporator (Doo Young High Technology, Seoul, Republic of Korea). Maltodextrin, in an amount equal to the solid content of the concentrate, was then added, and the mixture was stirred at 95 °C for 1 h. Finally, the concentrate was dried into a powder using a hot air dryer and stored at −20 °C until use. To ensure the quality control of the SBP extract, the active compound content was analyzed and specifications were established using high-performance liquid chromatography (HPLC, Waters, Milford, MA, USA). Additionally, the Korea Health Functional Food Association (Seongnam, Republic of Korea), an accredited testing organization, conducted tests on the SBP extract for properties such as Organoleptic properties, nutritional components, heavy metal content, *E. coli* contamination, total aflatoxin levels, and pesticide residues, confirming that no abnormalities were present and ensuring the safety of the extract. Furthermore, the moisture content of the powder was measured to verify its stability during storage, and the SBP extract was subsequently used in this study.

### 2.3. Study Design

The entire study from participant recruitment to the end of the trial lasted 8 months, from April to December. A total of 76 subjects with allergic rhinitis symptoms between April and November were enrolled in this study to evaluate the anti-allergic effects of SBP. However, based on the inclusion and exclusion criteria, 12 subjects were excluded, and 64 subjects were randomly assigned to the placebo and SBP groups (placebo, 32; SBP, 32). After randomization, the placebo group possessed a higher proportion (63%) of participants recruited between September and November when allergic rhinitis symptoms were the most severe, whereas the SBP group possessed a higher proportion (56%) of participants recruited between June and August when the symptoms tended to be milder. During the clinical trial period, eight participants withdrew for reasons such as withdrawal of consent (1 in SBP group), not following protocol (1 in SBP group, 3 in Placebo group), use of contraindicated drugs (2 in SBP group), and COVID-19 (1 in SBP group), and 56 participants (placebo = 29, SBP = 27) completed the test (Figure 1). Before participating in the clinical trial, the recruited participants or their representatives were provided a thorough explanation of the trial. If they voluntarily agreed to participate, they signed a consent form, and the necessary checks and examinations were conducted at that point. The actual clinical trial, excluding the screening phase, was conducted over 6 weeks. Any tests that were not completed during visit 1 could be performed during a revisit within the screening period. Visit 2 took place within 28 days of visit 1, during which random assignment and prescription of the SBP or placebo were carried out, and the necessary items were examined. Visit 3 occurred 3 weeks after starting SBP or placebo, and visit 4 marked the end of the human application trial that took place 6 weeks after starting SBP or placebo. The examination items corresponding to each time point are presented in Figure 2.

### 2.4. Primary Indicators

The Reflective Total Nasal Symptoms Score (rTNSS) evaluates symptoms of runny nose, itchy nose, nasal congestion, sneezing, and postnasal drip on a 4-point scale, where 0 = no symptoms, 1 = mild symptoms, 2 = moderate symptoms, and 3 = most severe symptoms. The total rTNSS was calculated as the sum of the four symptom scores at each visit.

### 2.5. Secondary Indicators

For the secondary indicators, all scores for instantaneous TNSS (iTNSS) symptoms that appeared on the day of the visit (visits 2, 3, and 4) were recorded and presented as results. To assess the quality of life related to allergic rhinitis, the Rhinoconjunctivitis Quality of Life Questionnaire (RQLQ) survey was conducted. The survey consisted of seven domains (activity, sleep, general symptoms, daily activities, nasal symptoms, eye symptoms, and emotions) and measured the degree of discomfort of the symptoms on a 7-point scale, where 0 = not troubled, 1 = mildly troubled, 2 = minimally troubled, 3 = moderately troubled, 4 = much troubled, 5 = very troubled, and 6 = extremely troubled. Finally, the Global Impression of Change (PGIC) assessment, where patients evaluate their overall health status themselves, was used to calculate the degree of improvement on a scale from −3 to +3 points, where −3 = very worse, −2 = much worse, −1 = minimal worse, 0 = no change, 1 = minimal improved, 2 = much improved, and 3 = very improved.

### 2.6. Tertiary Indicators

Blood samples were collected at each visit (visits 2, 3, and 4) to measure pathological markers after an 8 h fast. The pathological markers that were evaluated included cytokines (TNF-α, IL-4, and IL-5), IgE, ECP, histamine, prostaglandin, and leukotrienes. Blood samples were stored frozen until analysis of cytokines (TNF-α, IL-4, and IL-5), histamine, prostaglandin, and leukotriene and analyzed simultaneously after the last visit of all subjects.

### 2.7. Cell Culture

EoL-1 cells (DSMZ, Braunschweig, Germany), a human eosinophilic leukemia cell line, were cultured at a density of 1 × 10^6^ cells/mL in Roswell Park Memorial Institute (RPMI) 1640 medium (Thermofisher scientific, Waltham, MA, USA) with 10% fetal bovine serum (GenDepot, Katy, TX, USA) and incubated at 37 °C in a 5% CO_2_ environment. Differentiation was induced by the addition of 0.5 mM butyrate (Sigma–Aldrich, St. Louis, MO, USA) for 2 days. And then, 1 ng/mL lipopolysaccharide (Sigma–Aldrich) was treated for 24 h to produce a series of inflammatory mediators associated with various immune responses.

The human mast cell line HMC-1 obtained from Merck Millipore (Darmstadt, Germany) was maintained in Iscove’s Modified Dulbecco’s Medium (IMDM) (Thermofisher scientific) supplemented with 10% fetal bovine serum, 1.2 mM α-thioglycerol (Sigma–Aldrich), and 100 U/mL penicillin/streptomycin sulfate (Thermofisher scientific). Cells were incubated at 37 °C in a humidified atmosphere containing 5% CO_2_.

### 2.8. Real-Time PCR Analysis

Total RNA was isolated and purified using the RNeasy Mini Kit (Qiagen, Hilden, Germany). Following this, RNA was quantified using a NanoDrop™ 8000 Spectrophotometer (Thermofisher scientific), and cDNA was synthesized utilizing the Reverse Transcription System (Promega, Madison, WI, USA). Real-time PCR was performed using SYBR Green (GenDepot) and a real-time PCR detection system (Qiagen). The primer sequences are listed in Table 1.

### 2.9. Western-Blot Analysis

EoL-1 cells differentiated with butyrate were cultured for 24 h with or without LPS and SBP treatment, and the cells were lysed on ice for 40 min with lysis buffer. Lysed cells were centrifuged at 4 °C, 12,000× *g* for 20 min, and the supernatant was collected to quantify proteins according to the bicinchoninic acid assay (GenDepot). Then, proteins were separated using 4–20% sodium dodecyl–sulfate polyacrylamide gel electrophoresis (SDS-PAGE), transferred to polyvinylidene fluoride (PVDF) membranes (Bio-Rad, Hercules, CA, USA), and blocked with 5% skim milk for 1 h at room temperature. The membrane was incubated overnight at 4°C with primary antibodies PI3K (abcam, Cambridge, MA, USA, ab22653), p-PI3K (Cell Signaling Technology, Danvers, MA, USA; 4228), AKT (Cell Signaling Technology, 9272), p-AKT (Cell Signaling Technology, 4056), mTOR (Cell Signaling Technology, 2972), p-mTOR (Cell Signaling Technology, 5536), and β-actin (Cell Signaling Technology, 4967) and then incubated with HRP-conjugated secondary antibodies for 1 h at room temperature. Antibody binding was visualized using chemiluminescence reagent (Thermofisher scientific), and the signal was detected using a Chemi-doc image detector (Bio-Rad) and ImageJ software (ver. 1.8.0.172, National Institutes of Health).

### 2.10. Analysis of ECP Production in EOL-1 Cells

Culture supernatants were collected from butyrate- and LPS-treated EoL-1 cells, with or without SBP. The collected supernatant was centrifuged at 2000× *g* for 5 min to remove debris and then used in the experiment. The amount of ECP in the cell supernatant was measured using an ELISA kit according to the manufacturer’s protocol (Abcam).

### 2.11. Analysis of Inflammatory Cytokine and Allergic Mediator Production in HMC-1 Cells

EoL-1 cells were seeded in a 12-well plate and incubated for 24 h. Upon reaching 85–90% confluence, the cells were treated with butyrate to induce differentiation for 2 days. Following differentiation, LPS and SBP were added, and the cells were further incubated for 24 h. The culture supernatant was collected and added to HMC-1 cells, which were cultured for an additional 24 h. The cultured supernatant was collected after centrifugation for 5 min at 2000× *g* to remove debris. The levels of inflammatory cytokine IL-4 (ab215089; Abcam), IL-5 (ab215536, Abcam), IFN-γ (ab46025; Abcam), and the allergic mediators histamine (ab213975, Abcam) and leukotriene B4 (LTB_4_, ab133040; Abcam) were quantified using enzyme-linked immunosorbent assay (ELISA) according to the manufacturers’ protocols. Target molecule concentrations in the supernatants were determined using standard curves generated from serial dilutions of standards.

### 2.12. Statistical Analysis

Descriptive statistics for the participants’ general characteristics and study variables were calculated as frequencies, percentages, means, and standard deviations. A *t*-test (or Mann–Whitney test) was performed to compare the means between the SBP group and the placebo group (clinical study) or the control group (in vitro experiment), and a chi-square test was used to examine the statistical differences in proportions between the groups. To test the mean differences in health status variables over time, repeated-measures ANOVA was performed. Statistical significance was set at *p* < 0.05. All statistical analyses were performed using SPSS (ver. 25) and R (ver. 4.1.2).

## 3. Results

### 3.1. General Information

As a result of collecting the demographic characteristics of the clinical trial subjects, no significant differences in age, sex, or height were observed between the SBP and placebo groups. The SBP group exhibited a significantly higher weight (68.9 ± 15.2 kg) compared to that of the placebo group (60.4 ± 13.1 kg) (*p* = 0.017). However, as weight was determined to be unrelated to rhinitis, it was not used as a covariate. Seven participants in the SBP group and eleven in the placebo group experienced preexisting medical conditions. These conditions included chronic and benign diseases such as diabetes, hypertension, dyslipidemia, and arrhythmia, but they were controlled and did not meet the exclusion criteria for the human application test. Additionally, there were no significant differences between the SBP and placebo groups in demographic baseline examination parameters such as systolic blood pressure, diastolic blood pressure, and pulse rate (Table 2).

### 3.2. Primary Outcomes

#### rTNSS

Regarding the change in the primary indicators (Total rTNSS), after 6 weeks of intake compared to baseline, the placebo group exhibited a decrease of 2.64 ± 2.04 points, and the SBP group exhibited a decrease of 3.94 ± 2.33 points, indicating a significant improvement in symptoms (*p* = 0.03) (Figure 3A). The change in each rTNSS symptom indicated that all symptoms, including runny nose, itchy nose, nasal congestion, sneezing, and postnasal drip, decreased in the SBP group after six weeks of intake compared to baseline, indicating an improvement in each rTNSS symptom. Particularly, the change in nasal congestion symptoms was 1.05 ± 0.64 points in the SBP group and 0.58 ± 0.60 points in the placebo group after 6 weeks of intake compared to baseline, highlighting a significantly greater decrease in the SBP group compared to that of the placebo group, indicating a significant improvement in symptoms (*p* = 0.007). Additionally, there was a significant difference in the symptoms between the groups depending on the intake period, confirming that the symptoms in the SBP group improved (interaction *p* = 0.012) (Figure 3B).

### 3.3. Secondary Outcomes

#### 3.3.1. iTNSS

For the change in iTNSS according to the intake period, the SBP group (4.22 ± 2.82) showed significantly greater improvement compared to the placebo group (2.79 ± 2.18) after 6 weeks of intake, indicating a significant difference in the iTNSS change between the two groups (*p* = 0.038) (Figure 4A).

Specifically, iTNSS nasal symptoms (runny nose, nasal itching, nasal congestion, sneezing, and postnasal drip) exhibited significant improvement in the SBP group compared to baseline starting from week 3 of supplementation. In particular, after 3 and 6 weeks of supplementation, the SBP group experienced a more than two-fold decrease in nasal congestion scores (3 weeks: 0.74 ± 0.94; 6 weeks: 1.00 ± 0.83) compared to those of the placebo group (3 weeks: 0.28 ± 0.75; 6 weeks: 0.48 ± 0.69), indicating significant symptom improvement in the SBP group (3 weeks *p* = 0.046, 6 weeks *p* = 0.014). A repeated-measures analysis also confirmed a significant improvement in nasal congestion symptoms in the SBP group (interaction *p* = 0.031) (Figure 4B).

#### 3.3.2. RQLQ

To evaluate the impact of allergic rhinitis on quality of life, an RQLQ survey was conducted. As a result of measuring symptom changes across the seven areas of the RQLQ, the activity scores were changed more than twice in the SBP group (3.48 ± 4.00 points) compared to the placebo group (1.52 ± 2.57 points) after 3 weeks of supplementation with placebo or SBP compared to baseline. This indicates that activity-related discomfort was significantly improved in the SBP group (*p* = 0.036) (Figure 5A). Additionally, compared to baseline, the SBP group exhibited significant improvements in discomfort related to regular, social, and outdoor activities after three weeks of consumption. In particular, discomfort in social activity, which exhibited similar levels between the groups at baseline (placebo group: 2.41 ± 1.09 points, SBP group: 2.85 ± 1.46 points), decreased by 0.48 ± 0.95 points in the placebo group after 3 weeks of consumption, while it decreased by 1.22 ± 1.53 points in the SBP group, exhibiting more than twice the reduction and a significant improvement in social activity discomfort (*p* = 0.033) (Figure 5B).

At the baseline, the sleep symptom scores were 6.83 ± 4.17 points for the placebo group and 8.89 ± 3.47 points for the SBP group. After 3 weeks of supplementation, the SBP group experienced a significantly greater improvement in sleep quality, with a reduction of 4.11 ± 4.35 points compared to a reduction of 1.17 ± 2.90 points in the placebo group, indicating more than a threefold improvement (*p* = 0.005). Additionally, after 6 weeks, the sleep score in the placebo group decreased by 2.69 ± 3.38 points, while the SBP group exhibited a more substantial reduction of 5.37 ± 4.81 points, confirming a significant improvement in sleep quality (*p* = 0.019). The repeated-measures analysis further revealed a significant difference between the groups (interaction *p* = 0.004), indicating that sleep quality in the SBP group progressively improved over the supplementation period (Figure 5C). The improvement in the average change in sleep quality in the SBP group was attributed to the significant alleviation of symptoms, such as insomnia, tossing and turning, and sleep deprivation. In particular, after 6 weeks of consumption, the insomnia score decreased by 0.97 ± 1.02 points in the placebo group, while the SBP group exhibited a larger reduction of 1.81 ± 1.75 points, indicating a significant improvement in sleep quality (*p* = 0.03). The repeated-measures analysis also revealed a significant difference between the groups (*p* = 0.003), confirming the SBP group’s improvement in insomnia over the duration of test food consumption. Regarding another sleep evaluation parameter (the change in sleep deprivation scores), the SBP group (3 weeks: 1.48 ± 1.70; 6 weeks: 2.00 ± 1.59) exhibited a significant reduction compared to the placebo group (3 weeks: 0.34 ± 1.04; 6 weeks: 0.83 ± 1.31) after both 3 and 6 weeks of consumption (3 weeks *p* = 0.006, 6 weeks *p* = 0.004). Additionally, the repeated-measures analysis indicated a significant difference between the groups (*p* = 0.002), further supporting the effectiveness of the test food for improving sleep deprivation in the SBP group (Figure 5D).

As a result of evaluating the changes in systemic symptoms, the SBP group (3 weeks: 10.22 ± 8.16; 6 weeks: 12.67 ± 7.40) showed a significant change in symptom scores compared to the placebo group (3 weeks: 4.45 ± 6.56; 6 weeks: 7.14 ± 7.02) after 3 and 6 weeks of intake (3 weeks *p* = 0.005, 6 weeks *p* = 0.006), and a significant difference in the change in systemic symptoms according to repeated measurements was confirmed between the groups (interaction *p* = 0.002), confirming the SBP group’s effect of improving systemic symptoms (Figure 5E). The evaluation of systemic symptoms assessed seven categories that included fatigue, thirst, amount of work, lethargy, concentration, headache, and exhaustion. As a result, the SBP group showed significant improvement in all systemic symptom items after 3 weeks of intake compared to baseline. In particular, after 6 weeks of intake, the fatigue (2.26 ± 1.26), thirsty (1.93 ± 1.11), lethargic (2.15 ± 1.35), and concentration (2.07 ± 1.44) scores in the SBP group improved more than those in the placebo group (fatigue: 0.90 ± 1.18; thirsty: 0.79 ± 1.29; lethargic: 1.21 ± 1.42; concentration: 1.03 ± 1.45), confirming a significant difference between the groups (fatigue, *p* = 0.000; thirsty, *p* = 0.002; lethargic, *p* = 0.014; concentration, *p* = 0.01). In addition, repeated-measures analysis showed a significant difference in in the changes between the two groups for fatigue (interaction *p* = 0.000), thirsty (interaction *p* = 0.003), lethargic (interaction *p* = 0.008), and concentration (interaction *p* = 0.006).

As a result of evaluating the change in daily life, the SBP group decreased by 5.74 ± 3.84 points after 6 weeks of intake compared to the baseline, and the placebo group decreased by 3.62 ± 3.28 points, revealing a significant improvement in daily life symptoms in the SBP group (*p* = 0.030). Additionally, the change in daily life according to repeated measurements exhibited a significant difference between the groups (interaction *p* = 0.047), indicating that the SBP group exhibited a significant improvement in daily life depending on the period of intake of the test food (Figure 5G). In the daily life symptom evaluation, the SBP group exhibited a significant reduction in the frequency of tissues use (for runny nose and watery eyes), nose/eye rubbing, and repeated nose blowing compared to baseline after 3 weeks of consuming the test substance. After six weeks, the change in symptoms was more pronounced in the SBP group. Specifically, the reduction in the frequency of tissues use and repeated nose blowing were 1.10 ± 1.29 and 1.24 ± 1.41 points in the placebo group and 1.85 ± 1.38 and 1.78 ± 1.42 points in the SBP group, respectively. For both symptoms, the SBP group demonstrated a significantly greater improvement than that of the placebo group (*p* = 0.041 and *p* = 0.036, respectively) (Figure 5H).

As a result of the RQLQ nasal symptom change evaluation for the improvement of nasal symptoms, the SBP group decreased by 4.67 ± 4.87 points after 3 weeks of intake and by 6.85 ± 4.49 points after 6 weeks compared to the baseline, and the placebo group decreased by 1.97 ± 3.61 points after 3 weeks of intake and 3.31 ± 3.27 points after 6 weeks. The SBP group exhibited a significant improvement in nasal symptoms after 3 (*p* = 0.021) and 6 weeks (*p* = 0.001). Additionally, repeated measurements indicated a significant difference in nasal symptom changes between the SBP and placebo groups (interaction *p* = 0.002), confirming a statistically significant improvement in nasal symptoms in the SBP group (Figure 5I). The evaluation of nasal symptoms included assessment of nasal congestion, runny nose, sneezing, and postnasal drip. After 6 weeks of consumption, the change in nasal congestion symptoms decreased by 0.79 ± 1.11 points in the placebo group but exhibited a significantly larger reduction of 1.93 ± 1.33 points in the SBP group, confirming a notable improvement in symptoms in the SBP group (*p* = 0.001). The repeated-measures analysis also demonstrated a significant difference in symptom changes between the groups, further confirming the improvement in nasal congestion in the SBP group (interaction *p* = 0.004). Regarding runny nose symptoms, after 6 weeks of consumption, the placebo group exhibited a reduction of 0.97 ± 1.24 points, while the SBP group exhibited a greater reduction of 1.85 ± 1.51 points, indicating a significantly greater improvement in the SBP group compared to that in the placebo group (*p* = 0.02). The repeated-measures analysis also indicated a significant difference between the groups, confirming that runny nose symptoms in the SBP group were effectively improved during the test substance consumption period (interaction *p* = 0.035). For sneezing symptoms, after 6 weeks, the placebo group exhibited a reduction of 0.97 ± 1.21 points, and the SBP group exhibited a reduction of 1.67 ± 1.21 points. A greater reduction in the SBP group led to significant symptom improvement after 6 weeks (*p* = 0.035). The repeated-measures analysis also demonstrated a significant improvement in sneezing symptoms in the SBP group (interaction *p* = 0.043). For postnasal drip symptoms, an improvement in symptoms was observed in the SBP group after 3 weeks. After 6 weeks, the placebo group exhibited a reduction of 0.59 ± 1.05 points, and the SBP group exhibited a reduction of 1.41 ± 1.53 points. A greater reduction in the SBP group led to significant symptom improvement after 6 weeks (*p* = 0.022). The repeated-measures analysis also confirmed a significant difference between the groups, indicating that postnasal drip symptoms effectively improved in the SBP group (interaction *p* = 0.014) (Figure 5J). Overall, the evaluation scores for nasal symptoms in the SBP group were significantly different from those in the placebo group after six weeks of consumption. The repeated-measures analysis confirmed that nasal symptoms, including nasal congestion, runny nose, sneezing, and postnasal drip, effectively improved in the SBP group during the test substance consumption period.

As a result of the eye symptom change evaluation, it was confirmed that the change in eye symptom scores in the SBP group significantly improved compared to the placebo group after 3 weeks of intake compared to the baseline (*p* = 0.016) (Figure 5K). The eye symptoms in the SBP group, including itching, wateriness, soreness, and swelling, exhibited significant improvement at 3 weeks after consumption compared to baseline. In particular, for the wateriness symptom, the placebo group exhibited a change of 0.34 ± 0.81 points from 2.14 ± 1.81 points at baseline to 1.79 ± 1.82 points after 3 weeks, and the SBP group showed a change of 0.90 ± 1.40 points from 2.37 ± 1.67 points at baseline to 1.11 ± 1.15 points after 3 weeks. It was confirmed that the wateriness symptoms were significantly improved by showing a greater decrease than the placebo group with a point decrease (*p* = 0.009). Additionally, the repeated-measures analysis revealed a significant difference between the groups, further confirming meaningful symptom improvement in the SBP group (interaction *p* = 0.036). Regarding sore eyes, after 3 weeks, the placebo group exhibited a reduction of 0.48 ± 1.27 points compared to baseline, while the SBP group exhibited a more pronounced reduction of 1.33 ± 1.33 points. This over two-fold decrease in the SBP group compared to the value for the placebo group reflects a significant improvement in eye soreness symptoms (*p* = 0.018) (Figure 5L).

When evaluating the improvement in emotion, the placebo group decreased by 1.97 ± 3.20 points after 3 weeks of intake and by 3.72 ± 3.99 points after 6 weeks compared to the baseline, and the SBP group decreased by 4.70 ± 4.54 points after 3 weeks of intake and by 6.11 ± 3.78 points after 6 weeks. Accordingly, the SBP group exhibited a significant improvement in emotional symptoms after 3 weeks of intake (*p* = 0.011) and at 6 weeks (*p* = 0.026). Additionally, repeated measurements indicated a significant difference in the degree of change in emotional symptoms between the groups (*p* = 0.018), and the SBP group exhibited a significant improvement in emotional symptoms (Figure 5M). Additionally, the SBP group exhibited improvements in emotional symptoms, such as frustration, tiredness, irritation, and embarrassment, with all emotions significantly decreasing from 3 weeks after the start of consumption compared to baseline. In particular, the SBP group exhibited notable improvement in patience and embarrassment. Patience exhibited a decrease of 0.62 ± 1.29 points from 1.52 ± 1.27 points at baseline to 0.90 ± 1.05 points after 6 weeks in the placebo group, while in the SBP group, it exhibited a more substantial decrease of 1.30 ± 1.20 points from 1.85 ± 1.29 points at baseline to 0.56 ± 0.89 points after 6 weeks. This greater reduction in the SBP group indicates a significant improvement in patience-related emotional symptoms after 6 weeks of consumption (*p* = 0.048). For embarrassment, the changes in symptoms were more pronounced in the SBP group. The placebo group exhibited a decrease of 0.41 ± 0.95 points after 3 weeks and of 1.14 ± 0.99 points after 6 weeks, while the SBP group exhibited a decrease of 1.41 ± 1.53 points after 3 weeks and of 1.78 ± 1.25 points after 6 weeks. This greater reduction in the SBP group revealed a significant improvement in embarrassment symptoms (*p* = 0.005 at 3 weeks and *p* = 0.038 at 6 weeks). Furthermore, the repeated-measures analysis confirmed significant differences between the groups, with the SBP group exhibiting a significant improvement in emotional symptoms related to embarrassment (interaction *p* = 0.006). This confirmed the effect of the test substance on reducing feelings of embarrassment (Figure 5N).

Therefore, the SBP group that consumed the test food for 6 weeks exhibited significant improvements in sleep quality, systemic symptoms, daily life, nasal symptoms, and emotional symptoms compared to those of the placebo group, indicating that the SBP group experienced an improvement in the quality of life during allergic rhinitis.

After evaluating the seven areas of the RQLQ according to the intake period in the SBP and placebo groups, the symptom scores for all areas were combined and presented as a result (Figure 6). At baseline, the total RQLQ scores were 64.83 ± 30.32 points in the placebo group and 74.70 ± 25.63 in the SBP group. After 3 and 6 weeks of supplementation, the SBP group (3 weeks: 36.07 ± 29.90 points; 6 weeks: 48.04 ± 28.12 points) exhibited a significantly greater reduction in total RQLQ scores compared to that of the placebo group (3 weeks: 15.21 ± 20.00 points; 6 weeks: 28.45 ± 22.59 points), indicating a significant improvement in symptoms (3 weeks *p* = 0.004, 6 weeks *p* = 0.006). Additionally, the repeated-measures analysis indicated a significant difference between the groups, confirming significant symptom improvement in the SBP group (interaction *p* = 0.001).

#### 3.3.3. PGIC

As a result of evaluating the effect of the test food on PGIC, the subjects who were evaluated as having improved after 3 and 6 weeks of intake were 55.18% and 79.31% in the placebo group and 77.78% and 81.48% in the SBP group, respectively. The PGIC significantly improved depending on the duration of test food intake (interaction *p* = 0.004), confirming that SBP was effective for alleviating allergic rhinitis symptoms (Figure 7).

### 3.4. Tertiary Outcomes

#### Inflammatory Cytokine and Allergic-Mediated Factors

As a result of evaluating the effect of the test food on the change in cytokine concentration, the placebo group exhibited an increase in TNF-α concentration after 3 and 6 weeks compared to baseline, whereas the SBP group exhibited a significant decrease in TNF-α after 6 weeks of intake compared to baseline (*p* < 0.004). The IL-4 concentration did not change after consuming the test food compared to baseline in either the placebo or SBP groups. The IL-5 concentration did not change after 6 weeks of consumption of the test food compared to baseline in the placebo group, whereas the SBP group exhibited a significant decrease after 6 weeks of intake compared to baseline (*p* < 0.036).

Additionally, when an allergic reaction is initiated with a protein produced and secreted by eosinophils, the production amount of ECP that is proportional to the degree of eosinophil activity decreased by 0.12 ± 2.90 ng/mL in the placebo group and by 2.97 ± 6.86 ng/mL in the SBP group from baseline to 6 weeks of intake, exhibiting a significant decrease in the SBP group (*p* < 0.0001). A significant difference between the groups was also confirmed (*p* = 0.005). These results confirmed that the test food SBP significantly reduced the concentration of ECP and effectively improved the inflammatory response in allergic rhinitis subgroups. The concentrations of the allergy mediators IgE and histamine were measured at baseline, 3 weeks after ingestion, and 6 weeks after ingestion. However, no significant improvements in blood levels were observed following the administration of either placebo or SBP. Changes in the concentrations of prostaglandins and leukotrienes that are other inflammatory substances secreted from mast cells during an allergic reaction increased in the placebo group after 3 and 6 weeks compared to baseline, whereas in the SBP group, the concentrations tended to decrease after 6 weeks of intake compared to baseline (Table 3).

### 3.5. Effect of SBP on Eosinophil Cationic Protein mRNA Expression and Secretion in EoL-1 Cells Induced by Butyrate

Eosinophil cationic protein (ECP) is a major protein present in the granules of eosinophils, which are white blood cells that play a crucial role in immune responses to parasitic infections and in the pathophysiology of allergic diseases such as asthma [36,37]. ECP levels increase in response to allergens and contribute to the symptoms of allergic rhinitis, such as nasal congestion and mucus production [38,39]. Therefore, we evaluated the efficacy of SBP for reducing ECP levels in the human eosinophil cell line EoL-1. Previous studies have reported that EoL-1 cells can differentiate into mature eosinophil-like cells upon treatment with butyrate, resulting in a significant increase in the expression of CCR3, secondary eosinophil granules (ECP and EDN), and IL-5Ra [40]. In our study, EoL-1 cells that differentiated into mature eosinophils following butyrate treatment exhibited increased ECP gene expression and secretion, and treatment with SBP led to a concentration-dependent decrease in ECP levels (Figure 8).

### 3.6. Effect of SBP on Inflammatory Cytokine mRNA Expression and Secretion in HMC-1 Cells Stimulated with Culture Supernatants of EoL-1 Cells

EOL-1 cells differentiated into mature eosinophils by butyrate were treated with SBP, and the culture supernatant was collected. The supernatant was then treated with LPS and HMC-1, and the gene expression and secretion levels of cytokines were measured. As presented in Figure 9, treatment with the EOL-1 supernatant and LPS resulted in increased expression and secretion of the pro-inflammatory cytokines IL-4 and IL-5, while the anti-inflammatory cytokine IFN-γ was decreased compared to levels in the placebo group. When cells in this inflammatory state were treated with SBP, the levels of pro- and anti-inflammatory cytokines were significantly regulated, demonstrating anti-inflammatory efficacy.

### 3.7. Effect of SBP on Allergic Mediator Secretion in HMC-1 Cells Stimulated with Culture Supernatants of EoL-1 Cells

After treating BA-differentiated EOL-1 cells with SBP and culturing, the supernatant was collected and added to HMC-1 cells. Subsequently, LPS was administered, and the secretion levels of the allergy mediators LTB_4_ and histamine were measured, with the results presented in Figure 10. The results indicated that unlike the placebo group where BA and LPS stimulation increased the production of allergy mediators, treatment with SBP led to a reduction in these levels. Notably, at a concentration of 200 μg/mL, a significant decrease in both LTB_4_ and histamine was observed.

### 3.8. Effect of SBP on PI3K/Akt/mTOR Signaling in EoL-1 Cells

To further analyze the molecular mechanisms underlying eosinophil activation and degranulation that regulate ECP activity, we investigated the expression of components of the PI3K/Akt/mTOR-signaling pathway. The results indicated that in EOL-1 cells differentiated with BA and LPS, there was an induction of phosphorylation of PI3K, Akt, and mTOR. However, SBP treatment significantly reduced the levels of phosphorylated proteins. Overall, although BA and LPS stimulation promoted the phosphorylation of PI3K/Akt/mTOR in EOL-1 cells, SBP treatment inhibited this phosphorylation (Figure 11, Appendix A). This suggests that the regulation of eosinophil activation and degranulation, both of which are key markers of inflammatory and allergic responses, is mediated by the PI3K/Akt/mTOR-signaling pathway.

## 4. Discussion

SBP has been reported as a natural functional material with proven anti-allergic effects, primarily through the inhibition of the NF-κB-signaling pathway [28]. This suppression leads to a reduction in pro-inflammatory substances and inhibits the activation of the PI3K/mTOR pathway, and this is crucial for asthma treatment [21]. SBP exerts its anti-allergic effects by modulating the differentiation of Th1 and Th2 cells. Given that the role of SBP in reducing inflammation and alleviating allergic symptoms has been supported by both cellular and animal studies, it possesses promising potential for development as a health-promoting food ingredient.

Building on scientific evidence of the anti-allergic efficacy of SBP, our study aimed to evaluate the effects of SBP on alleviating allergic rhinitis symptoms in individuals suffering from discomfort and reduced quality of life due to this condition. Over a 6-week period, SBP and placebo were administered, with the goal of SBP reducing allergic symptoms. The evaluation tools included the rTNSS, iTNSS, RQLQ, PGIC, and immune activity biomarkers.

The rTNSS questionnaire that quantifies the severity of nasal symptoms such as sneezing, runny nose, nasal congestion, itching, and postnasal drip has proven to be a useful tool for monitoring the clinical efficacy of rhinitis treatment. Additionally, the RQLQ that is widely used in clinical research and practice evaluates the severity of daily life impairment and nasal and ocular symptoms in subjects with allergic rhinitis across seven domains and 28 questions. Developed according to well-established principles for chronic disease assessment tools, the RQLQ can effectively detect significant clinical changes, even over short periods of time [41,42].

Results from this study indicated that after 6 weeks of supplementation, the SBP group exhibited a significant improvement in nasal congestion symptoms as measured by rTNSS compared to that of the placebo group (*p* = 0.012). The SBP group also exhibited significant improvements in the overall RQLQ scores (*p* = 0.001) and in specific domains, such as sleep (*p* = 0.004), systemic symptoms (*p* = 0.002), daily activities (*p* = 0.047), nasal symptoms (*p* = 0.002), and emotions (*p* = 0.018). Symptom-specific assessments revealed significant improvements in insomnia (*p* = 0.003), lack of sleep (*p* = 0.002), fatigue (*p* = 0.000), thirst (*p* = 0.003), lethargy (*p* = 0.008), and concentration (*p* = 0.006) in the SBP group. Additionally, the SBP group exhibited greater reductions in nasal congestion (*p* = 0.004), runny nose (*p* = 0.035), sneezing (*p* = 0.043), and postnasal drip (*p* = 0.014) than did the placebo group, with significant reductions in emotional stress due to nasal symptoms (*p* = 0.006). Correspondingly, the overall improvement in the RQLQ scores was reflected in the PGIC evaluations, with the SBP group exhibiting greater improvement than that of the placebo group (*p* = 0.004). For most evaluation variables, symptom improvement was observed in the placebo group, similar to that in the SBP group. However, this improvement was not attributed to the placebo intake itself but rather to seasonal factors, as the recruitment for the placebo group was concentrated in September. This is due to the observation that while approximately 1 million people suffer from allergic rhinitis throughout the year, the highest number of cases occurs in September (approximately 1.15 million) and October (approximately 1.16 million) as summer transitions to fall [5]. Therefore, it was expected that the placebo group that possessed a higher proportion of participants recruited in September and October would exhibit higher baseline symptom scores, and symptom improvement over time could be attributed to natural seasonal factors.

Furthermore, the third efficacy variable, immune-related biomarkers, indicated significant reductions in the levels of the inflammatory cytokines TNF-α, PGE, LTB4, and ECP in the SBP group after 6 weeks of supplementation. While the placebo group exhibited an increase in these markers, the SBP group experienced a decrease, particularly in ECP levels that are crucial for diagnosing allergic and inflammatory conditions. Reduction in these markers is associated with the regulation of the eosinophil function that plays a multifaceted role as an immune response regulator in allergic inflammation [7,43,44].

To further explore the mechanisms underlying these effects, we investigated the PI3K/Akt/mTOR-signaling pathway in human eosinophil and mast cell models. PI3K/Akt/mTOR pathway not only serves as a critical regulator of eosinophil survival, activation, and degranulation but also controls eosinophil function and ECP release by regulating the balance between Th1/Th2 cells and cytokine secretion, thereby reducing tissue damage and alleviating symptoms of inflammatory and allergic diseases. Investigating this pathway is essential for elucidating its contribution to eosinophil-mediated immune responses and the pathogenesis of allergic and inflammatory diseases. Based on this theory, human eosinophilic leukemia cells (EoL-1) serve as an ideal model for in vitro studies to explore the relationship between this pathway and AR [45,46]. However, under normal conditions, EoL-1 cells exhibit morphological characteristics similar to myeloblasts, with a limited number of granules containing eosinophil peroxidase and a typical bilobed nucleus [45]. Therefore, differentiation of EoL-1 cells is necessary to conduct studies related to allergic inflammation and immune responses. EoL-1 cells can be differentiated into mature eosinophil-like cells using agents such as butyrate (BA), dibutyryl cyclic adenosine monophosphate (dbcAMP), interferon-gamma (IFN-γ), or phorbol 12-myristate 13-acetate (PMA) [47,48]. Among these, BA is the most commonly used agent, which activates Toll-like receptors (TLRs) upon treatment, enhancing the expression of TLR3, TLR4, and TLR7, while promoting nuclear segmentation, an increased cytoplasmic-to-nuclear ratio, and granule formation [40,47,49,50]. Furthermore, BA-differentiated EoL-1 cells treated with lipopolysaccharide (LPS), a TLR ligand, show a significant upregulation of transcription factors involved in Th2 cytokine production, which drives eosinophil differentiation [40]. The combination of BA and LPS treatment thus facilitates the differentiation of EoL-1 cells into mature eosinophil-like cells, providing a valuable in vitro model for evaluating the regulation of inflammatory mediators and immune responses. Previous studies have reported that this pathway regulates eosinophilic inflammation and tissue remodeling in CRSwNP by partially controlling autophagy levels [12]. In asthma, activation of the PI3K/Akt/mTOR pathway promotes the proliferation of airway smooth muscle cells and airway epithelial cells, leading to airway thickening and narrowing, and modulation of Th2 cell proliferation, activation, and apoptosis that are central to the inflammatory environment [15]. Notably, studies have demonstrated that Akt influences Th cell differentiation and cytokine production, disrupting the Th1/Th2 balance and contributing to the pathogenesis [51]. Increased PI3K and Akt activity has been observed in asthma models, and PI3K inhibitors have been demonstrated to suppress specific pathological symptoms, such as reduced chemokine expression in eosinophils, lower IL-5 and IL-13 levels in bronchoalveolar lavage fluid, reduced eosinophil infiltration in lung tissue, and decreased airway hyperresponsiveness [52,53]. Additionally, inflammatory cytokines secreted by activated Th2 cells stimulate B cells to produce IgE, which binds to receptors on mast cells, leading to the release of allergic mediators such as histamine and leukotrienes, thereby triggering allergic reactions [54]. Similar findings were observed in our study, where SBP regulated the production of IL-4, IL-5 (Th2 cytokines), and IFN-γ (Th1 cytokine) in EOL-1 and HMC-1 cells, ultimately reducing ECP release and the production of histamine, LTB4, and PGE through Th1/Th2 balance regulation and eosinophil degranulation prevention.

Finally, no significant differences in vital side and adverse effects were observed between SBP group placebo group intake during the trial, ensuring the safety of SBP. Many natural materials and compounds have potential therapeutic effects, but there are only a limited number of cases in which efficacy and safety have been verified through scientific evaluation. In this study, we investigated in vitro, in vivo, and clinical trials to determine whether SBP significantly alleviates allergic rhinitis symptoms. By demonstrating safety while showing an effect, it has been suggested that it can be developed as a functional food. In particular, unlike existing synthetic antihistamines, SBP is a natural product-based bioactive substance with a low risk of side effects when taken long-term and has an immune-boosting effect. It has the potential to contribute to the management of various allergic diseases through regulation. This study establishes the usability of SBP as a functional raw material and provides scientific evidence that can contribute to the development of natural product-based allergy treatment and the development of the functional food market.

## 5. Conclusions

This study evaluated the effects and safety of SBP on allergic rhinitis symptoms and the immune function in individuals experiencing discomfort due to these conditions. The results demonstrated that SBP significantly improved nasal congestion, as measured by rTNSS. Additionally, as part of the secondary efficacy outcomes, SBP led to significant improvements in the quality of life (RQLQ) related to sleep disturbances, systemic symptoms (such as fatigue and concentration), daily activities, nasal symptoms (nasal congestion, runny nose, sneezing, and postnasal drip), and emotions. The overall improvement in the subjects, as assessed by the PGIC, was also significant in the SBP group. Moreover, SBP significantly reduced the ECP levels, a biomarker of allergic and inflammatory conditions, in a tertiary efficacy evaluation study. This reduction indicated that SBP effectively alleviated allergic and inflammatory responses. This study further confirmed that SBP modulates ECP activity through the PI3K/Akt/mTOR-signaling pathway, thereby suppressing the activation and degranulation of eosinophils and mitigating allergic rhinitis symptoms and related hyperimmune responses. In conclusion, SBP improved the key indicators of allergic and inflammatory responses, confirming its functional efficacy and safety for enhancing immune function and alleviating allergic rhinitis symptoms. Building on these findings, the proven safety of SBP highlights its potential as a promising functional ingredient for managing allergic rhinitis, and if developed as a health functional food in the future, it has the prospect of contributing to improved health and quality of life for patients.

## Figures and Tables

**Figure 1 nutrients-17-00468-f001:**
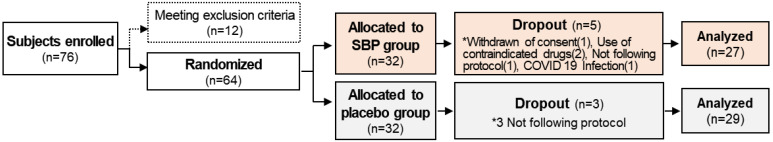
Research flow chart. The diagram indicates the flow of the subjects after enrolment and randomization in the placebo or the SBP group. * Reasons for the dropout of the subject.

**Figure 2 nutrients-17-00468-f002:**
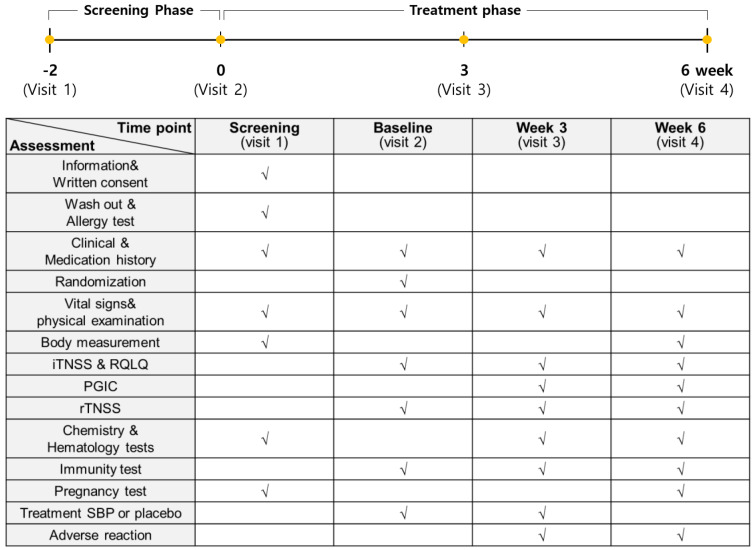
Study assessments conducted during each stage. The examination items at each time point are marked with “√”.

**Figure 3 nutrients-17-00468-f003:**
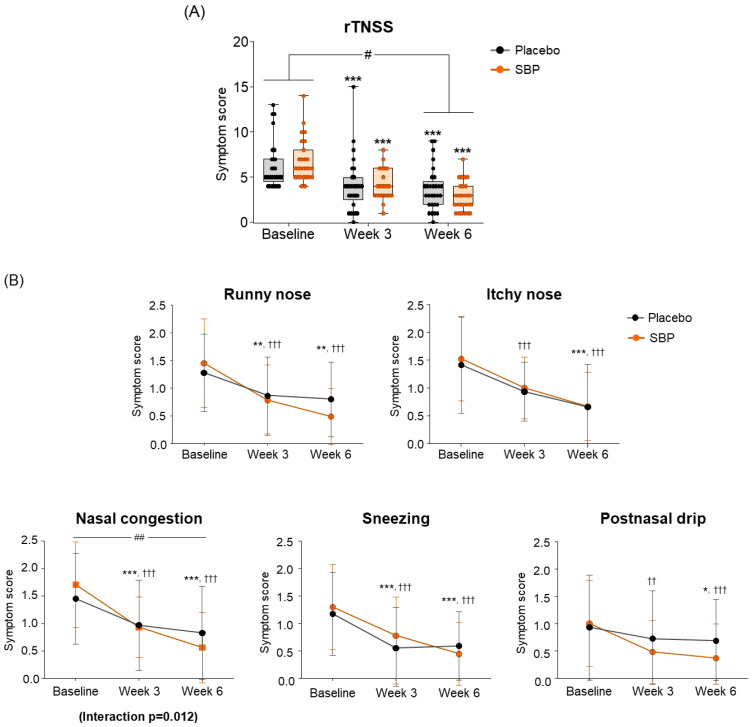
Effect of SBP on reflective total nasal symptoms (rTNSS) and specific symptoms improvement. (**A**) rTNSS average score and (**B**) each symptom scores according to placebo or SBP intake. rTNSS survey evaluated symptoms on 4-point scale, where 0 = no symptoms, 1 = mild symptoms, 2 = moderate symptoms, and 3 = most severe symptoms. Values are mean ± SD. Significant difference compared to baseline for the placebo group (* *p* < 0.05, ** *p* < 0.01, *** *p* < 0.001); significant difference compared to baseline for the SBP group (†† *p* < 0.01, ††† *p* < 0.001); significant difference between groups from baseline to week 6 (# *p* < 0.05, ## *p* < 0.01); Interaction *p* = interaction between groups and time.

**Figure 4 nutrients-17-00468-f004:**
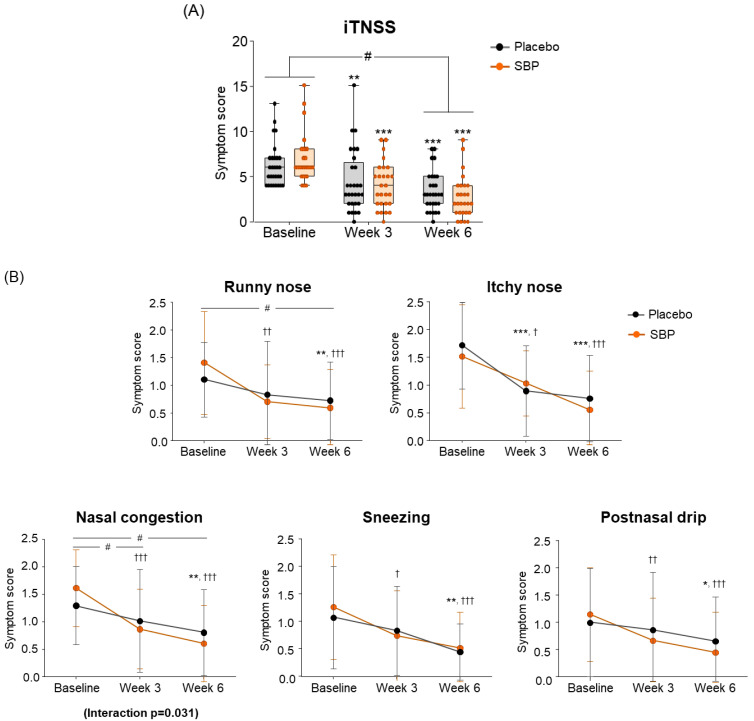
Effect of SBP on instantaneous TNSS (iTNSS) and specific symptoms improvement. (**A**) iTNSS average score and (**B**) individual symptom scores according to placebo or SBP intake. iTNSS survey evaluated symptoms on 4-point scale, where 0 = no symptoms, 1 = mild symptoms, 2 = moderate symptoms, and 3 = most severe symptoms. Values are mean ± SD. Significant difference compared to baseline for the placebo group (* *p* < 0.05, ** *p* < 0.01, *** *p* < 0.001); significant difference compared to baseline for SBP group († *p* < 0.05, †† *p* < 0.01, ††† *p* < 0.001); significant difference between groups from baseline to Week 3 or 6 (# *p* < 0.05); interaction *p* = interaction between groups and time.

**Figure 5 nutrients-17-00468-f005:**
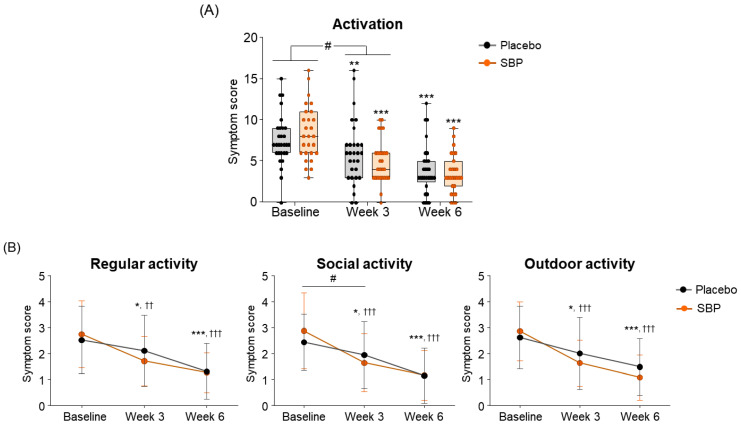
Effect of SBP on activation and specific symptom improvement in the Rhinoconjunctivitis Quality of Life Questionnaire (RQLQ). (**A**–**N**) Changes in each symptom of RQLQ scores according to placebo or SBP intake. The RQLQ survey evaluated symptoms on a 7-point scale, where 0 = not troubled, 1 = mildly troubled, 2 = minimally troubled, 3 = moderately troubled, 4 = much troubled, 5 = very troubled, and 6 = extremely troubled. Values are mean ± SD. Significant difference compared to baseline for the placebo group (* *p* < 0.05, ** *p* < 0.01, *** *p* < 0.001); significant difference compared to baseline for the SBP group (†† *p* < 0.01, ††† *p* < 0.001); significant difference between groups from baseline to week 3 or 6 (# *p* < 0.05, ## *p* < 0.01, ### *p* < 0.001).

**Figure 6 nutrients-17-00468-f006:**
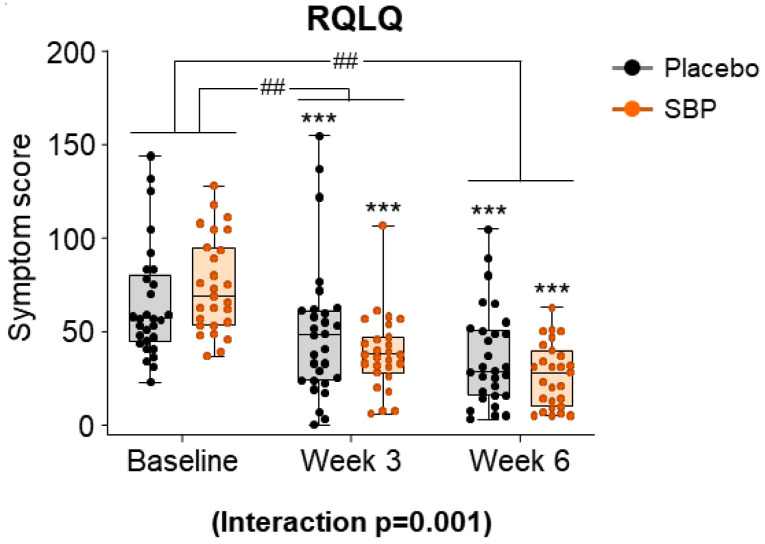
Effect of SBP Rhinoconjunctivitis Quality of Life Questionnaire (RQLQ) on improvement. The RQLQ survey evaluated symptoms on a 7-point scale, where 0 = not troubled, 1 = mildly troubled, 2 = minimally troubled, 3 = moderately troubled, 4 = much troubled, 5 = very troubled, and 6 = extremely troubled. Values are mean ± SD. Significant difference compared to baseline for the placebo group (*** *p* < 0.001); significant difference between groups from baseline to week 3 or 6 (## *p* < 0.01); interaction *p* = interaction between groups and time.

**Figure 7 nutrients-17-00468-f007:**
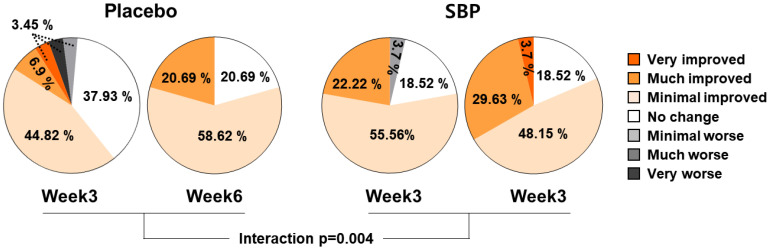
PGIC changes according to placebo or SBP intake. Interaction *p* = interaction between groups and time.

**Figure 8 nutrients-17-00468-f008:**
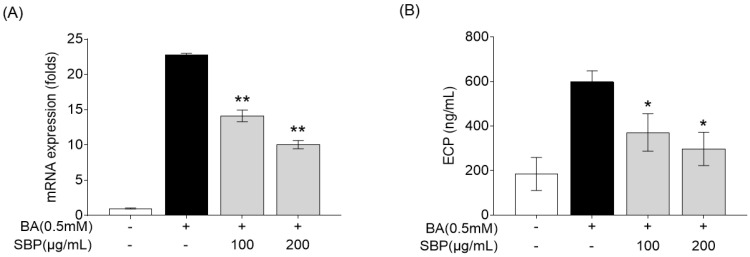
Effects of SBP extracts on ECP in the differentiation of EoL-1 cells. EOL-1 cells were induced to differentiate by adding BA for 2 days, followed by treatment with butyrate and SBP for 24 h. Subsequently, the cells or culture supernatants were used to measure (**A**) the relative mRNA expression level determined by qPCR and (**B**) the production as determined by ELISA. Data are expressed as mean ± SD. *p*-value compared to the BA-only-treated group according to Student’s *t*-test (* *p* < 0.05, ** *p* < 0.01). SBP, sword bean pod; BA, butyrate.

**Figure 9 nutrients-17-00468-f009:**
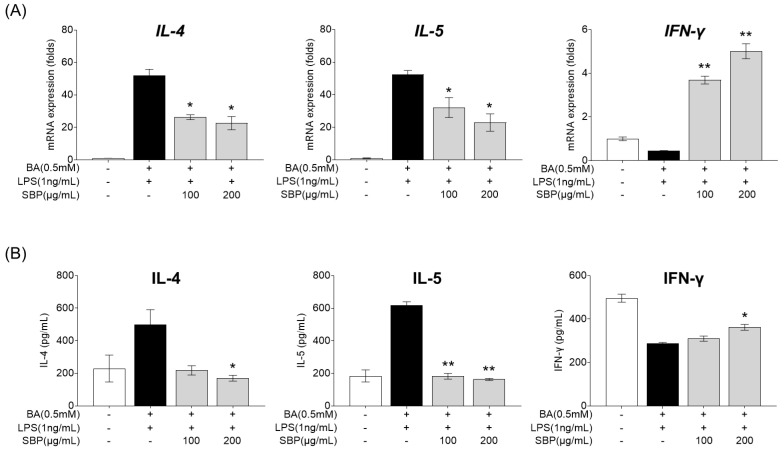
Effects of SBP extracts on inflammatory cytokines in HMC-1 cells. EOL-1 cells were induced to differentiate by BA for 2 days, and then treated with or without LPS, and SBP for 24 h. After treatment, the EOL-1 cell culture supernatants were used to stimulate the HMC-1 cells for 24 h. Inflammatory cytokine levels were measured by (**A**) relative mRNA expression determined by qPCR and (**B**) cytokine production determined by ELISA. Data are expressed as mean ± SD. *p*-value compared to the BA/LPS-treated group according to Student’s *t*-test (* *p* < 0.05, ** *p* < 0.01). SBP, sword bean pod; BA, butyrate; LPS, lipopolysaccharide.

**Figure 10 nutrients-17-00468-f010:**
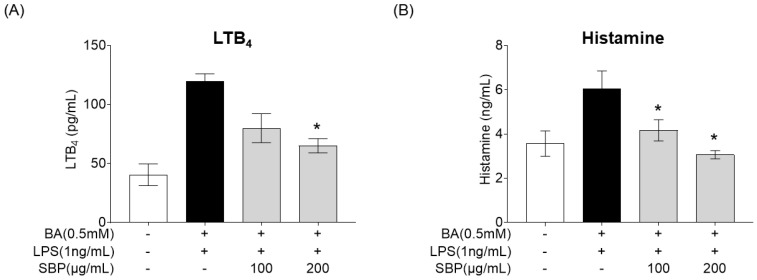
Effects of SBP extracts on allergic mediator secretion in HMC-1 cells. EoL-1 cells were induced to differentiate by BA for 2 days, and then treated with or without LPS and SBP for 24 h. After treatment, the EOL-1 cell culture supernatants were used to stimulate the HMC-1 cells for 24 h. (**A**) Leukotriene B_4_ and (**B**) histamine production were measured by ELISA. Data are expressed as mean ± SD. *p*-value compared to the BA/LPS-treated group by Student’s *t*-test (* *p* < 0.05). SBP, sword bean pod; BA, butyrate; LPS, lipopolysaccharide, LTB_4_, leukotriene B4.

**Figure 11 nutrients-17-00468-f011:**
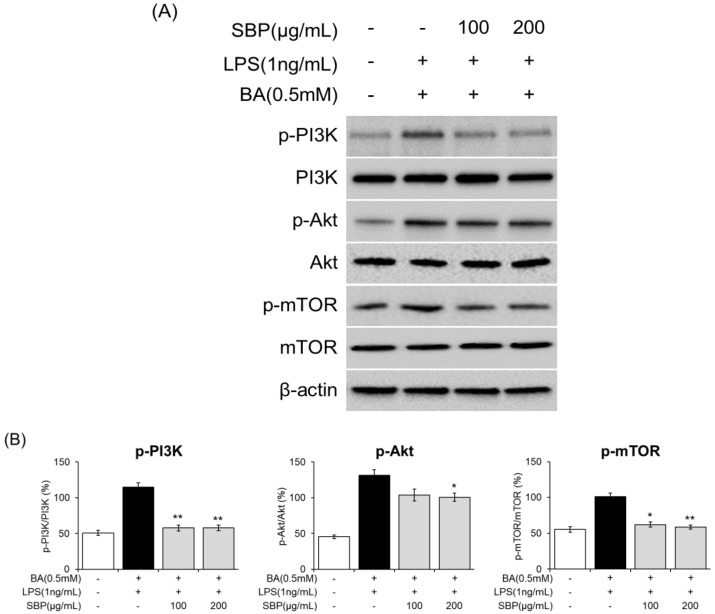
Effects of SBP extracts on the expression of PI3K/Akt/mTOR-signaling molecules in EOL-1 cells. EOL-1 cells were induced to differentiate by adding BA for 2 days, followed by treatment with LPS and SBP for 24 h. For PI3K/Akt/mTOR-signaling molecules, (**A**) protein levels were measured by Western blot, and (**B**) densitometry was quantified by Image J. Data are expressed as mean ± SD. *p*-value compared to the BA/LPS-treated group according to Student’s *t*-test (* *p* < 0.05, ** *p* < 0.01). SBP, sword bean pod; BA, butyrate; LPS, lipopolysaccharide.

**Table 1 nutrients-17-00468-t001:** Primer sequences.

Gene	Primer Sequences (5′→3′)	
*ECP*	Forward	ATAGTTTTCACCCAGAGTCCA
	Reverse	TGCCCGCATTGCAATGGTGCATCGA
*IL-4*	Forward	AACGGCTCGACAGGAACCT
	Reverse	ACTCTGGTTGGCTTCCTTCCA
*IL-5*	Forward	GCTTCTGCATTTGAGTTTGCTAGCT
	Reverse	TGGCCGTCAATGTATTTCTTTATTAAG
*IFN-* *γ*	Forward	TCAGCTCTGCATCGTTTTGG
	Reverse	GTTCCATTATCCGCTACATCTGAA
*GAPDH*	Forward	CTGGGCTACACTGAGCACC
	Reverse	AAGTGGTCGTTGAGGGCAATG

**Table 2 nutrients-17-00468-t002:** Comparison of baseline characteristics by placebo and SBP groups.

Variables	Treatment Group			*p*-Value
	Total (n = 64)	SBP (n = 32)	Placebo (n = 32)	
Age (years)	38.3 ± 11.8	38.3 ± 11.9	38.4 ± 12	0.803
Sex, n (%)				0.214
Male	13.0 ± 20.3	9.0 ± 28.1	4.0 ± 12.5	
Female	51.0 ± 79.7	23.0 ± 71.9	28.0 ± 87.5	
Height (cm)	163.8 ± 8.5	165.9 ± 9.8	161.8 ± 6.4	0.093
Body weight (kg)	64.7 ± 14.7	68.9 ± 15.2	60.4 ± 13.1	0.017
BMI (kg/m^2^)	24.0 ± 4.3	24.8 ± 3.9	23.2 ± 4.5	0.076
Systolic blood pressure (mmHg)	124.0 ± 14.0	126.4 ± 15.7	121.5 ± 12.1	0.166
Diastolic blood pressure (mmHg)	75.0 ± 10.8	76.7 ± 11.3	73.3 ± 10.2	0.220
Pulse rate (n/min)	81.1 ± 11.5	81.9 ± 11.9	80.2 ± 11.3	0.562
Medical history, n (%)				0.258
N	47.0 ± 73.4	26.0 ± 81.2	21.0 ± 65.6	
Y	17.0 ± 26.6	6.0 ± 18.8	11.0 ± 34.4	
Concomitant drug, n (%)				0.109
N	57.0 ± 89.1	31.0 ± 96.9	26.0 ± 81.2	
Y	7.0 ± 10.9	1.0 ± 3.1	6.0 ± 18.8	
Physical examination, n (%)				
Normal	64.0 ± 100.0	32.0 ± 100.0	32.0 ± 100.0	
Abnormal	0.0 ± 0.0	0.0 ± 0.0	0.0 ± 0.0	

**Table 3 nutrients-17-00468-t003:** Change in inflammatory cytokine and allergic mediated factors from baseline.

	Placebo				SBP			
	Baseline	Week 3	Week 6	Change	Baseline	Week 3	Week 6	Change
TNF-α (pg/mL)	4.19 ± 1.52	4.52 ± 1.72	4.41 ± 1.76	−0.22 ± 1.44	4.59 ± 1.85	5.00 ± 3.66	4.30 ± 2.01	0.29 ± 0.89 *
IL-5 (pg/mL)	0.15 ± 0.04	0.20 ± 0.15	0.14 ± 0.04	0.01 ± 0.03 *	0.18 ± 0.09	0.18 ± 0.09	0.16 ± 0.05	0.02 ± 0.07 *
ECP (ng/mL)	4.60 ± 3.25	4.98 ± 3.89	4.48 ± 2.56	0.12 ± 2.90	10.20 ± 10.16	9.55 ± 8.23	7.23 ± 5.22	2.97 ± 6.86 *^,#^
Prostaglandin (pg/mL)	856.17 ± 03.09	944.29 ± 573.18	932.20 ± 729.28	−76.03 ± 254.42 *	744.97 ± 529.52	709.48 ± 389.18	714.86 ± 462.25	30.12 ± 446.28
Leukotriene (pg/mL)	98.14 ± 45.24	117.74 ± 122.70	107.17 ± 80.35	−9.03 ± 77.95	136.05 ± 104.42	140.34 ± 112.25	126.94 ± 110.57	9.11 ± 78.88

Values are mean ± SD. Change represents the difference between baseline and week 6. Significant differences compared to baseline within the same group (* *p* < 0.05); significant differences between groups (# *p* < 0.05).

## Data Availability

The original contributions presented in this study are included in the article/Appendix A. Further inquiries can be directed to the corresponding authors.

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
