# Peer review of "Immature Sword Bean (Canavalia gladiata) Pod Alleviates Allergic Rhinitis (A Double-Blind Trial) Through PI3K/Akt/mTOR Signaling"

_nutrients, 2025, doi:10.3390/nu17030468_

Round 1
Reviewer 1 Report
Comments and Suggestions for Authors
The investigations conducted by Hwang et al. deserve to be considered for publication in Nutrients after the following revisions:
Clarify your study’s objectives in the abstract. In this section, the results and the values with statistical significance have to be highlighted.
The study’s aims have to be clearly stated at the end of the Introduction. Provide some illustrations in this section.
Line 105: Was this study approved on 22nd April 2005? If this is correct, can you please explain why you are only submitting your work to be published in 2025? This is 20 years later!
Can you please better explain the analysis of inflammatory cytokine and allergic mediator production in HMC-1 cells (section 2.9).
In my point of view subsection 3.1 should not be in Results and it should be moved to section 2. Only Table 2 should be retained in Section 3.
In the Results section, some parts should be summarized and more concise. Here, you only have to report your results.
After the Discussion, a new section should be included to mention your study’s limitations and strengths.
Improve your Conclusions by providing future perspectives and recommendations for further investigations.
Author Response
Response to Reviewer 1 Comments
|
Summary |
|
|
Thank you very much for taking the time to review this manuscript. According to the reviewer's comments, we respond point-by-point to the comments and suggestions of the reviewer as follows and have edited the re-submitted files.
|
|
Comments 1: Clarify your study’s objectives in the abstract. In this section, the results and the values with statistical significance have to be highlighted.
→ According to the reviewer's comments, we explained the research objectives in detail and presented statistical values for meaningful results (Line 16-28 ; “Therefore, in this study, we aimed to evaluated the symptom relief efficacy …… PI3K/Akt/mTOR activation, resulting in decreased eosinophil activity.).
Comments 2: The study’s aims have to be clearly stated at the end of the Introduction. Provide some illustrations in this section.
→ According to the reviewer's comments, we have revised the research objectives in the introduction(Line 114-120 ; “Therefore, in this study investigated …… alleviating allergic rhinitis symptoms.”) and also presented with the graphical abstract.
Comments 3: Line 105: Was this study approved on 22nd April 2005? If this is correct, can you please explain why you are only submitting your work to be published in 2025? This is 20 years later!
→ According to the reviewer's comments, we revised it in order of date, month, and year according to the paper published in the Nutrients journal (Line 125 ; “IRB NO. DKUH 2021-12-014, Date. 05 April 2022”).
Comments 4: Can you please better explain the analysis of inflammatory cytokine and allergic mediator production in HMC-1 cells (section 2.9).
→ According to the reviewer's comments, we added detailed descriptions in the manuscript(section 2.9) regarding cell culture and differentiation methods, and quantification of inflammatory cytokines and allergic mediators. These contents have been revised on the line 271-284 of manuscript (“EoL-1 cells were seeded in a 12-well plate …… from serial dilutions of standards.”).
Comments 5: In my point of view subsection 3.1 should not be in Results and it should be moved to section 2. Only Table 2 should be retained in Section 3.
→ According to the reviewer's comments, The subject registration procedure and schematic diagram included in subsection 3.1 were moved to subsection 2.3 (Line 183-201 ; “The entire study from participant …… was conducted over 6 weeks.). And also, figure 2 revised to make it easier to understand.
Comments 6: In the Results section, some parts should be summarized and more concise. Here, you only have to report your results.
→ According to the reviewer's comments, we briefly summarized the secondary outcome results for iTNSS and RQLQ. These contents have been revised on the line 350-690 of manuscript (“The change in iTNSS according to the …… were more pronounced in the SBP group.”).
Comments 7: After the Discussion, a new section should be included to mention your study’s limitations and strengths.
→ This study is meaningful in that it scientifically demonstrates the allergy improvement efficacy and safety of SBP, a natural material, to verify its value as a functional material for health functional foods and pharmaceuticals, and this information was added on the line951-964 of manuscript (“Finally, no significant differences…… of the functional food market.”).
Comments 8: Improve your Conclusions by providing future perspectives and recommendations for further investigations.
→ SBP is an allergy-improving natural product functional material that can be consumed for a long time and reduce side effects, and is expected to be used as a raw material for developing health functional foods and pharmaceuticals. These contents were added on the line 984-987 of manuscript (“Building on these findings, …… health and quality of life for patients.”).

Reviewer 2 Report
Comments and Suggestions for Authors
This study reported the clinical effect of the Immature Sword Bean (Canavalia gladiata) Pod against Allergic Rhinitis.
1. Revise the first sentence of the abstract. Do not use "=".
2. "SBP treatment also led to a notable reduction in eosinophil cationic protein levels, a key biomarker of allergic inflammation, by inhibiting PI3K/Akt/mTOR activation, resulting in decreased eosinophil activity." They have clarify which findings are from cell models.
3. The quality control of the SBP extract have to be disclosed.
4. The exclusion criteria should be mentioned.
5. The reasons for the combination for modeling of LBS and / or BA should be explained.
6. Gene names should be italic.
7. Too many figures. Please combine some.
8. The rational of phosphorylation o PI3K/Akt/mTOR for AR treatment should be explained more clearly.
Author Response
|
Response to Reviewer 2 Comments
|
||
|
Summary |
|
|
|
Thank you very much for taking the time to review this manuscript. Please find the detailed responses below and the corresponding revisions/corrections highlighted/in track changes in the re-submitted files. |
||
|
|
||
|
Comments 1: Revise the first sentence of the abstract. Do not use "=". |
||
|
→ According to the reviewer’s comments, we deleted “=” of abstract (Line 14).
Comment 2: "SBP treatment also led to a notable reduction in eosinophil cationic protein levels, a key biomarker of allergic inflammation, by inhibiting PI3K/Akt/mTOR activation, resulting in decreased eosinophil activity." They have clarify which findings are from cell models. → SBP has been scientifically demonstrated to effectively improve allergic rhinitis by regulating key biomarkers of allergic inflammation through inhibition of PI3K/Akt/mTOR activation in EoL-1 and HMC-1 cells.
|
||
|
Comments 3: The quality control of the SBP extract have to be disclosed. |
||
|
→ According to the reviewer’s comments, we analyzed the active compound content, nutritional contents, heavy metal content, E. coli, pesticide residue, storage stability, etc. of the SBP extract to secure the standard specifications, safety, and stability of the extract and set quality control standards. These contents were added on the line 172-180 of manuscript (“To ensure the quality control …… was subsequently used in this study.“).
Comments 4: The exclusion criteria should be mentioned. → According to the reviewer’s comments, we added the exclusion criteria information in subsection 2.1 on Lines 140-160 (“The exclusion criteria were as follows: …… unsuitable by the investigator.”).
Comments 5: The reasons for the combination for modeling of LBS and / or BA should be explained. → The cell model to effectively simulate allergic inflammatory reactions and eosinophil activation was used in the experiment by inducing the secretion of inflammatory mediators (ECP, IL-5) and inflammatory cytokines (IL-1β, IL-6, TNF-α) through BA and LPS treatment in EoL-1 cells. These contents were added on the line 909-932 of manuscript (“PI3K/Akt/mTOR pathway not only…… mediators and immune responses.”).
Comments 6: Gene names should be italic. → We revised all genetic names in the manuscript to italics (Table 1 : ECP, IL-4, IL-5, IFN-γ, GAPDH, Figure 9 : IL-4, IL-5, IFN-γ).
Comments 7: Too many figures. Please combine some. → According to the reviewer's comments, we combined the Fig. 5-11 into Fig. 5. Continuing from then on the Figure numbers were revised on the line 449-845 of manuscript.
Comments 8: The rational of phosphorylation PI3K/Akt/mTOR for AR treatment should be explained more clearly. → Inhibition of phosphorylation of the PI3K/Akt/mTOR pathway reduces ECP secretion by inhibiting eosinophil activity and regulates the secretion of inflammatory cytokines by maintaining the balance of Th1/Th2 cells. Through this, we demonstrated that it improves key indicators of allergic rhinitis disease symptoms such as suppression of eosinophil infiltration in lung tissue and inhibition of airway smooth muscle cell contraction associated with rhinitis. The above is described in the manuscript on the Lines 82-93 (“In addition, the PI3K/Akt/mTOR pathway plays…… proliferation via the PI3K/mTOR pathway”). |
||

Round 2
Reviewer 2 Report
Comments and Suggestions for Authors
There is no further comment.
Author Response
We appreciate the reviewer’s valuable feedback regarding the insufficient details in the methodology. In response, we have revised the Methods section to provide a more comprehensive description of the Western blot analysis and densitometry procedures. Specifically, we have added details to method 2.8.